# STRIDE: A Tool-Assisted LLM Agent Framework for Strategic and Interactive Decision-Making

## Abstract

Large Language Models (LLMs) like GPT-4 have revolutionized natural language processing, showing remarkable linguistic proficiency and reasoning capabilities. However, their application in strategic multi-agent decision-making environments is hampered by significant limitations including poor mathematical reasoning, difficulty in following instructions, and a tendency to generate incorrect information. These deficiencies hinder their performance in strategic and interactive tasks that demand adherence to nuanced game rules, long-term planning, exploration in unknown environments, and anticipation of opponents' moves. To overcome these obstacles, this paper presents a novel LLM agent framework equipped with memory and specialized tools to enhance their strategic decision-making capabilities. We deploy the tools in a number of economically important environments, in particular bilateral bargaining and multi-agent and dynamic mechanism design. We employ quantitative metrics to assess the framework's performance in various strategic decision-making problems. Our findings show that our enhanced framework significantly improves strategic decision-making capability of LLMs. While we highlight the inherent limitations of current LLMs, we demonstrate the improvements through targeted enhancements, suggesting a promising direction for future developments in LLM applications for interactive environments.

## 1. Introduction

Large language models (LLMs) such as GPT-4 have demonstrated exceptional proficiency in generating coherent natural language from textual inputs (Bubeck et al., 2023). They display human-like strategic thinking and excel at flexible reasoning with nuanced, context-specific information (Aher et al., 2022; Kwon et al., 2023; Suzgun et al., 2022). These achievements have sparked significant interest in their potential for decision-making in complex environments (Yao et al., 2022; Shen et al., 2024; Wang et al., 2023).

To further integrate LLMs into our society, such as deploying them as fiduciary agents on behalf of individuals or organizations in a competitive environment where human and AI agents coexist, the ability to reason strategically is of vital importance. However, due to their inherent limitations in basic mathematics (Bubeck et al., 2023), instruction following (Jang et al., 2022), and susceptibility to hallucinations (Chen et al., 2023), the following challenges exist: (i) LLMs may fail to accurately interpret game rules and objectives expressed in natural language, e.g., form a well-defined utility function that reflects their preference over possible outcomes (Guo et al., 2023); (ii) LLMs are generally inept at long-horizon planning to maximize their utility, which is essential in scenarios where decisions have extended consequences (Huang et al., 2024); (iii) They exhibit poor capabilities in strategic exploration of unknown environments (Krishnamurthy et al., 2024), which hampers their ability to optimize decisions on unforeseen conditions; (iv) LLMs have limited capacity in anticipating opponents' moves and adapting their strategies accordingly (Park et al., 2024), which is crucial for any competitive interaction. These limitations collectively underscore the challenges in deploying LLMs for nuanced and dynamic strategic reasoning tasks.

In light of these challenges, this paper proposes *a LLM agent framework designed to enhance their **STR**ategic and **I**nteractive **DE**cision making capability, named STRIDE*, as illustrated in Figure 1. Compared to simply prompting the LLM with a description of the problem setup and potentially some chain-of-thought examples (Brookins & DeBacker, 2023; Gandhi et al., 2023), STRIDE can effectively address the aforementioned challenges and enhance the LLM's reasoning capability in strategic settings. Specifically, the LLM, which serves as the controller of the whole framework, orchestrates the reasoning process through a structured *Thought* sequence, as shown at the top of Figure 1. Compared with popular frameworks like ReAct (Yao et al.,

[1]Anonymous Institution, Anonymous City, Anonymous Region, Anonymous Country. Correspondence to: Anonymous Author <anon.email@domain.com>.

Preliminary work. Under review by the International Conference on Machine Learning (ICML). Do not distribute.

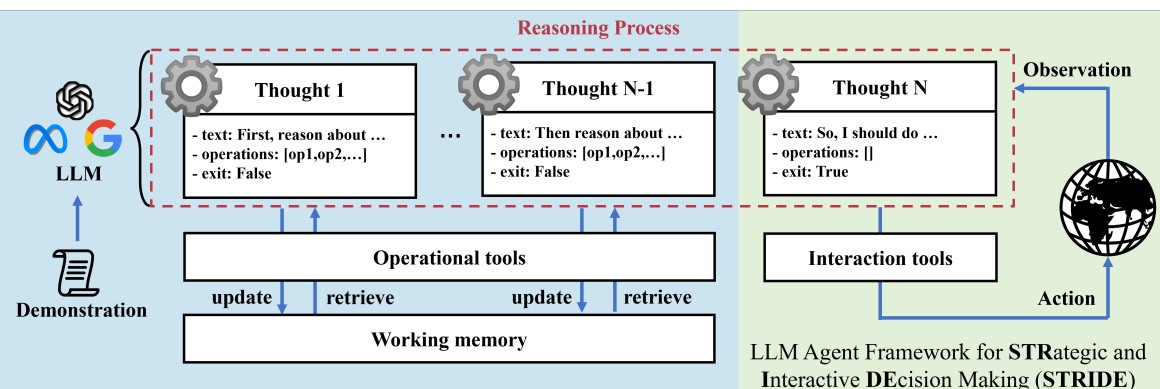

*Figure 1.* Illustration of STRIDE framework, which consists of a reasoning module powered by LLMs, a working memory that stores important parameters of the problem instance and intermediate results of the reasoning process, as well as tools that facilitate reasoning (taking care of low-level computation and managing the working memory) and acting (converting reasoning texts into executable actions).

2022) and Reflexion (Shinn et al., 2024), whose *Thought* step typically involves a single step of textual reasoning before interacting with the environment, which is depicted as the green region of Figure 1, our design is tailored to multi-step reasoning assisted with tools and memory, as shown in the blue part of Figure 1. Each *Thought* unit outlines a sequence of operations to be executed, which consist of tools specifically engineered to manage the low-level calculations needed in various decision-making scenarios. Additionally, an external working memory is integrated to preserve crucial parameters. Therefore, Challenge (i) can be addressed by executing an operation that evaluates the agent's utility in the *Thought* unit. Challenge (ii), which is mainly caused by the information loss in long-context (Liu et al., 2024), can be addressed by storing important problem parameters and intermediate results in the working memory. Challenges (iii) and (iv) can be addressed via a combination of demonstrations and dedicated operations that emuate the behavior of strategic exploration or belief update.

To evaluate our framework, we carefully choose a collection of decision-making problems that highlight the aforementioned challenges in significant and economically relevant real-world strategic settings. First, we evaluate our framework in a general single-agent Markov Decision Process (MDP), which exemplifies Challenges (ii) and (iii). Here the agent needs to strategically explore the unknown environment to improve their estimate of the transition and reward function, as well as planning over a long horizon to compute the optimal policy (Sutton & Barto, 2018). Second, we consider a dynamic mechanism design environment, which offers a multi-agent generalization of MDP where the mechanism designer seeks to maximize the cumulative sum of rewards over multiple agents based on agents' reported reward functions (Bergemann & Välimäki, 2010; 2019). In the multi-agent mechanism design environment, each agent has private information which evolves over time.

This problem covers Challenges (i)-(iv). The mechanism designer needs to anticipate the agents' strategic behavior and makes decisions, i.e., allocation and pricing rules, to ensure that truthfully reporting the reward function is unilaterally optimal for each agent. This setting encompasses many important allocation problems such as auctions for sponsored search and display advertising. Third, we consider an important class of bilateral bargaining games, where a seller and a buyer negotiate on the price of a good over a finite number of time steps under complete or incomplete information (Rubinstein, 1982; Fudenberg & Tirole, 1991). This covers Challenges (i), (ii) and (iv), as the agent needs to assess its utility for reaching a deal at different prices and time steps, inferring the opponent's private value based on his/her past responses, as well as predicting the opponent's future behaviors. The bargaining environment has many important applications in procurement and supply-chain sourcing decisions. For each strategic environment, we offer quantitative metrics that allow us to conclude whether the agent succeeded in making the optimal decisions based on the available information.

Through extensive empirical evaluation on these selected decision-making problems, we show that, with few demonstrations, the proposed framework can make strategic decisions on new problem instances with high success rate. This highlights the transformative potential of integrating LLMs with specialized tools, memory, and control structures to enhance strategic decision-making capabilities.

## 2. STRIDE LLM Agent Framework

In this section, we first present the building components of the STRIDE framework, explain how these components interact to support strategic decision-making, and then provide a detailed example by applying it to an autonomous driving scenario, which is modeled as a MDP.

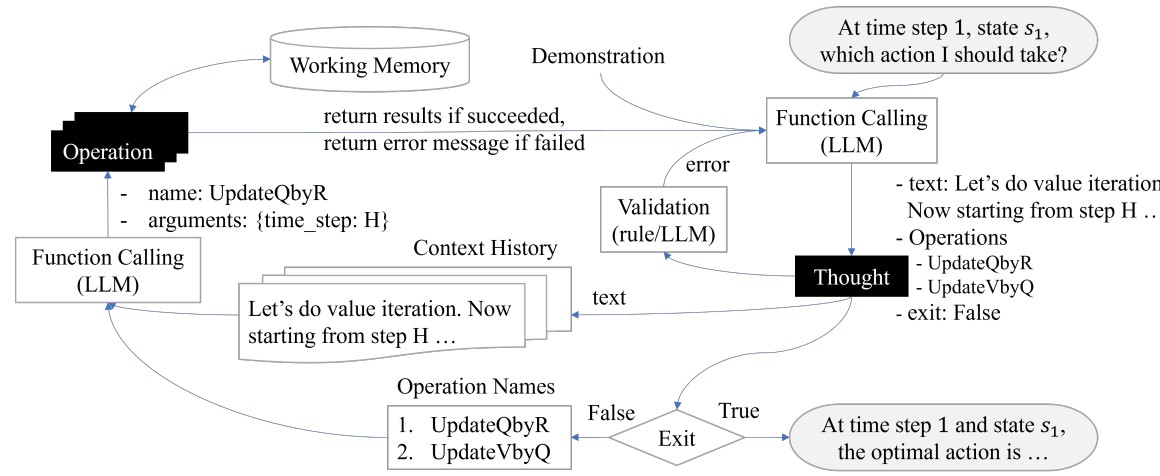

*Figure 2.* In STRIDE framework, the LLM controls the execution of operations and access to working memory via a sequence of *Thought* units. Each *Thought* unit is a structured data containing three fields: (i) text, which suggests the next step of strategic reasoning and summarize important information; (ii) operations: a list of operations to execute, in order to compute or retrieve information necessary for reasoning; (iii) exit: a boolean value indicating whether the reasoning process is completed. With proper demonstration and tools, STRIDE can emulate various algorithmic behaviors (value iteration algorithm here is one example) to facilitate strategic decision making.

## 2.1. Main Components of the STRIDE Framework

Our primary strategy to address the four challenges in Section 1 is to provide the LLM with tools taking care of low-level computation and a working memory retaining important parameters. Most importantly, we introduce a reasoning module that acts as the central executive, orchestrating the information flow among components and synthesizing structured *Thought* sequences to solve complex problems. Figure 2 provides a flowchart that illustrates how these components collaboratively facilitate strategic decision-making in an MDP. These components of STRIDE are introduced below.

**Tool Set.** As shown in Figure 1, the *tools* utilized by STRIDE are categorized into two distinct groups: operational tools, which are tailored to support sophisticated reasoning processes, and interaction tools, designed to enable STRIDE to interact effectively with its environment. What sets our work apart from previous LLM agents, such as ReAct (Yao et al., 2022) and Reflexion (Shinn et al., 2024), is the sophisticated integration of these operational tools by the *Thought* sequence to execute complex calculations that traditionally pose challenges for LLMs. For instance, these operations can calculate the utility of the agent based on the outcomes of a game or update the belief about uncertainty on the environment or the other agents. A combination of such operations allows STRIDE to implement various algorithmic behaviors such as dynamic programming to solve MDPs and Bargaining Games, facilitating a deeper and more precise decision-making process. They also let STRIDE scale to complex problems by abstracting detailed computations. This scalability is crucial in handling larger and more challenging scenarios that are beyond the capac-

ity of typical LLM agents. For instance, in the concrete example to be discussed later, STRIDE can easily handle a state space of size **120**. After completing the reasoning process, the resulting textual description of the LLM's decision is translated by the interaction tools into a structured format that is actionable within the specific environment, such as selecting an action in an MDP or offering a price in a sequential bargaining game.

**Reasoning Module.** To effectively leverage the operational tools for solving complex problems, we propose a unique design for the reasoning module, which is empowered by a pretrained LLM like GPT-4 (Achiam et al., 2023) or Claude (Anthropic, 2024), in the STRIDE framework. Using the MDP example in Figure 2, the reasoning process starts when the agent is prompted to answer the question about which action to take at the current time step, as shown on the top right corner. As the first step to answer this question, the LLM generates a *Thought* unit [1], whose text field describes a general plan about what needs to be done for the current reasoning step in order to answer the question and the operations field comprises an ordered list of operation names that the LLM deems necessary for completing the current step. For the MDP example in Figure 2, the LLM decides to use value iteration to compute the optimal policy, and thus the first step of its reasoning is to compute the Q values associated with the last time step $H$ (see Appendix B for details about value iteration). To do so, operations named `UpdateQbyR` and `UpdateVbyQ` are suggested by the LLM, as shown at the bottom of the figure. Note that

---

[1]This is done via the function calling feature of LLMs, which is commonly supported by models like GPT, Claude, and Gemini.

here the *Thought* unit only needs to specify which operations are necessary, that is, only the names are needed. The arguments for each selected operation are decided by the LLM on separate API calls based on the context history, as shown on the left of the figure. This particular design choice is motivated by our empirical observation that, letting the LLM simultaneously decide arguments for multiple operations is more prone to error.

Before the execution of the selected operations, the generated *Thought* unit undergoes a validation process based on predefined rules to ensure its integrity. For example, a common rule applied in our experiments is the mutual exclusivity of the exit condition and the presence of operations: the *Thought* unit must not simultaneously specify an exit as true while containing non-empty operations, as this often indicates a premature termination of the reasoning process. If this conflict occurs, the system will generate an appropriate error message, append it to the prompt, and prompt the LLM to revise and resubmit the *Thought* unit for re-evaluation. This mechanism ensures that operations proceed only with validated and logically consistent instructions. Enhanced applications of this functionality involve utilizing an additional LLM to verify whether the newly generated *Thought* unit adheres to the reasoning logic and language style presented in the demonstration. This step can improve consistency and prevent hallucinations. Employing a secondary LLM for cross-verification not only reinforces the accuracy of the *Thought* unit but also contributes to the ongoing research in maintaining coherence and reliability in LLM outputs, which is of independent interest. With the *Thought* unit validated, the selected operations will be executed in the specified order. The outcomes of these operations are then utilized to generate the subsequent *Thought* unit. This process continues until *Exit* is set to be true, signaling the completion of the reasoning process.

**Working Memory.** As mentioned in Section 1, for long-horizon planning, LLMs may forget or neglect important information mentioned early in the context history. Moreover, an accurate description of the problem instance sometimes require parameters of high dimensions, e.g., transition matrices of MDP. In this case, storing these parameters in the context history is costly and prone to error. Therefore, STRIDE is equipped with a working memory that stores these parameters, as well as intermediate results computed by the operational tools during the reasoning process. Information contains in the working memory is retrieved by the reasoning module for decision making.

## 2.2. Example: STRIDE Agent in Highway Environment

To illustrate the functionality of STRIDE, we apply it to the Highway Environment (Leurent, 2018) to optimally control a vehicle as depicted in Figure 3. We provide descriptions

of the operational tools constructed for the LLM and how the *Thought* sequence uses them for strategic reasoning.

**Tabular MDP Formulation.** This decision-making problem can be formulated as a tabular MDP with known transition function $P : \mathcal{S} \times \mathcal{A} \rightarrow \Delta(\mathcal{S})$ and reward function $R : \mathcal{S} \times \mathcal{A} \rightarrow \mathbb{R}$ ($\mathcal{S}$ and $\mathcal{A}$ denote the state and action spaces, respectively), where each state $s \in \mathcal{S}$ indexes a unique three-way tensor representing the time to collision with other vehicles[2], and the action set $\mathcal{A}$ includes changing to left lane, idle, changing to right lane, faster, and slower. Here we focus on a finite-horizon setting, i.e., the agent interacts with the environment for some fixed $H$ steps. At each step $h = 1, 2, \ldots, H$, the agent observes the current state $s_h \in \mathcal{S}$, and then chooses action $a_h \in \mathcal{A}$. The environment then produces a reward feedback $R(s_h, a_h)$ to the agent, with positive reward for staying in the right lane or maintaining a high speed, and negative reward for any collision, and then the state transits to $s_{h+1} \sim P(\cdot \mid s_h, a_h)$. It is known that the optimal policy, in this case, the fastest way to navigate through the traffic, can be computed using value iteration, which we will refer to as a reference algorithm for STRIDE. In the next paragraph, we show how to implement this algorithmic behavior in the reasoning of STRIDE with specialized operations and demonstration.

**Tools and *Thought* Sequence that Implement Value Iteration.** Value iteration starts from the end of the horizon $H$ and works backwards to the beginning, such that at each step $h \in [H]$, it updates the $Q_h(s, a) = R(s, a) + \sum_{s' \in \mathcal{S}} P(s'|s, a) V_{h+1}(s')$ and $V_h(s) = \max_{a \in \mathcal{A}} Q_h(s, a)$, with $V_{H+1}(s) = 0, \forall s$. During interaction, the agent can simply choose $a_h = \arg\max_{a \in \mathcal{A}} Q_h(s_h)$ for state $s_h$ at each step $h$. Therefore, to enable the LLM to compute the optimal policy for any MDP instance in this principled manner, we equip it with the following operations, i.e., a set of primitives that handle low-level calculations, thereby freeing the LLM to focus on higher-order reasoning.

- UpdateQbyR: add reward $R(s, a)$ to $Q_h(s, a)$ for all $(s, a)$ pairs at the specified time step $h$,
- UpdateQbyPV: add one-step look-ahead value $\sum_{s' \in \mathcal{S}} P(s'|s, a) V_{h+1}(s')$ to $Q_h(s, a)$ for all $(s, a)$ pairs at the specified time step $h$,
- UpdateV: take maximum $V_h(s) = \max_{a \in \mathcal{A}} Q_h(s, a)$ for the specified time step $h$,
- GetQ: retrieve the values of $Q_h(s, a)$ for all action $a \in \mathcal{A}$ at the specified time step $h$ and state $s$.
- GetArgMax: return the indices corresponding to the maximal value in the given list of numbers

To make the LLM generate *Thought* sequences that correctly utilize these operations to emulate value iteration, we

---

[2]for more details, see https://highway-env.farama.org/observations

*Figure 3.* Agent's objective in Highway Environment is to control the ego-vehicle, i.e., the green box, to reach a high speed while avoiding collision with the other vehicles, i.e., the blue boxes.

employ a structured demonstration generation approach:

- Implement value iteration algorithm using the provided operations to handle the computational intricacies;
- Augment the algorithm with annotated code that generates explanatory comments—the *Thought* text—at key steps to illustrate the logic and progression of the algorithm;
- Sample an instance of MDP, execute the augmented value iteration algorithm on this instance, and capture the resulting sequence of operation calls and textual comments.

With the generated demonstration, STRIDE not only performs calculations correctly but also maintains a logically coherent order when handling various MDP instances. Moreover, the flexible design of STRIDE as detailed in Figure 2 allows for emulating a broad spectrum of algorithms beyond value iteration. Therefore, STRIDE offers a flexible framework that can be extended to a diverse array of problem domains involving strategic decision-making, where algorithmic behavior of LLMs is critical. In particular, to tailor STRIDE to other domains, it suffices to construct domain-specific tool sets and provide demonstration to emulate other algorithms using these tools. As we will see in the sequel, STRIDE can be applied to dynamic mechanism design, two-player bargaining games, and zero-sum games such as Tic-Tac-Toe, where various tool sets and demonstration are constructed under the STRIDE framework.

## 3. Experiments

For each decision-making problem mentioned in Section 1, we first construct the operational tools and generate the corresponding demonstrations following the procedure described in Section 2.2, so that STRIDE is able to mimic the reference algorithm when solving each problem. Descriptions of the selected reference algorithms and the constructed operations can be found in Appendix B. Then to evaluate whether STRIDE can reliably solve new problem instances given provided demonstrations, we repeat experiments on randomly sampled instances and report the averaged results. For comparison, we include the following baseline agents: (i) *zero-shot Chain-of-Thought (CoT)*, (ii) *zero-shot CoT with code interpreter*, and (iii) *few-shot CoT with code interpreter*, where the latter two can utilize the coding capability of LLMs (through OpenAI Assistants API) to write and execute programs to solve the decision-making problems. Compared with (ii), (iii) is additionally provided with example implementation of the reference algorithm for each problem. Prompts used in all the experiments are given

in Appendix C. We also conducted additional experiments on other problem setups like Tic-Tac-Toe and Connect-N games to further demonstrate the generality of STRIDE. Details about these experiments are given in Appendix D.

### 3.1. Markov Decision Processes

We first evaluate STRIDE and the baselines (GPT-3.5-Turbo-0125 with temperature set to 0 is used for all agents) on MDP under both known model, where the transition function $P$ and reward function $R$ are given to the agent in the beginning, and unknown model, where the agent needs to estimate $P$ and $R$ during online interactions. In the following paragraphs, we first provide a formal definition of the objective of the agent under each setting, and then discuss the experiment setup and results.

**Agent's Objective in MDP with Known Model.** When the transition and reward functions are known to the agent, the objective is to find a policy, denoted as $\pi = \{\pi_h\}_{h=1}^{H}$ with $\pi_h : \mathcal{S} \to \Delta(\mathcal{A})$ for $h \in [H]$, that maximizes the expected cumulative rewards over $H$ time steps:

$$\max_\pi \mathbb{E}_{\pi,P}\left[\sum_{h=1}^{H} R(s_h, a_h)\right] := V_1^\pi, \qquad (1)$$

where the expectation is with respect to the randomness in state transitions and the stochasticity of $\pi$. Let's denote the optimal Q value function as $Q_h^\star(s,a)$ for $h \in [H]$. Then for any state $s_h$ encountered by the agent at step $h \in [H]$, we check whether the action $a_h$ taken by the agent satisfies $a_h = \arg\max_{a \in \mathcal{A}} Q_h^\star(s,a)$, and report the average success rate in the following experiment.

**Experiment Setup and Results.** We evaluate on MDPs with horizon length $H \in \{5, 10, 15\}$, number of states $|\mathcal{S}| \in \{3, 10\}$, and number of actions $|\mathcal{A}| \in \{3, 10\}$. For each configuration, we repeat the experiment for 20 times on randomly generated instances, by sampling dense tensors of size $\mathbb{R}^{S \times A \times S}$ and $\mathbb{R}^{S \times A}$ as the transition function and reward function, respectively. The average success rates are reported in Table 1. For STRIDE, we only provide it with a single demonstration that solves a MDP instance with $H = 5, S = 5, A = 5$. We can see that STRIDE outperforms the baselines in terms of finding the optimal policy for the given MDP instances.

**Agent's Objective in MDP with Unknown Model.** In this setting, $P$ and $R$ are unknown to the agent, but the agent is allowed to repetitively interact with the same MDP instance for a total number of $K$ episodes to explore and update its belief about $P$ and $R$ using the observed feedback. The

*Table 1.* Success rate in taking the optimal action (20 runs).

| H | S | A | zero-shot CoT | zero-shot CoT w/ code | few-shot CoT w/ code | STRIDE |
|---|---|---|---|---|---|---|
| 5 | 3 | 3 | 0.58 | 0.74 | 0.70 | **0.98** |
| 10 | 3 | 3 | 0.62 | 0.75 | 0.69 | **0.87** |
| 5 | 10 | 10 | 0.24 | 0.48 | 0.60 | **0.96** |
| 10 | 10 | 10 | 0.21 | 0.50 | 0.68 | **0.82** |

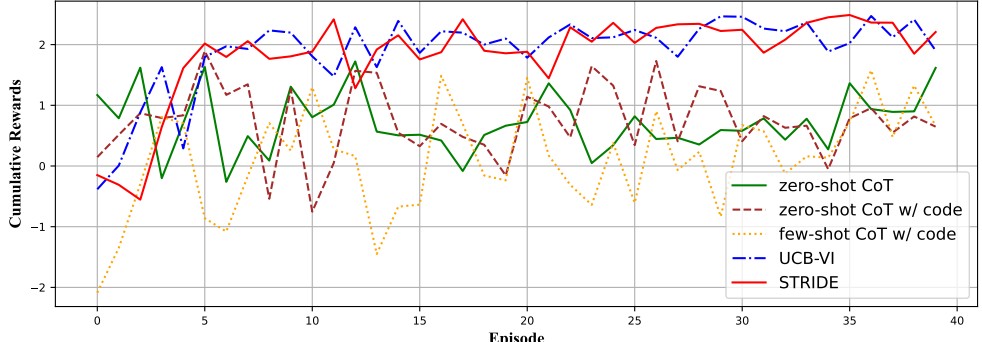

*Figure 4.* Comparison of cumulative rewards over episode. We observe that both STRIDE and UCB-VI exhibit rapid increases in their cumulative rewards, converging by approximately the 10-th episode. This indicates that STRIDE can effectively explore the environment, by emulating UCB-VI in its reasoning process. In contrast, the cumulative rewards of other baseline methods display ongoing fluctuations throughout the episodes, showing poor exploration ability in uncertain environments.

objective of the agent is to choose a sequence of policies $\pi^1, \pi^2, \ldots, \pi^K$, that minimizes the cumulative regret:

$$\min_{\pi^1, \pi^2, \ldots, \pi^K} \sum_{k=1}^{K} \left( V_1^{\pi^\star} - V_1^{\pi^k} \right). \tag{2}$$

In addition to the challenge of long-horizon planning exemplified by Eq (1), Eq (2) also requires addressing the exploration-exploitation dilemma. Specifically, the agent needs to strategically balance between exploring unfamiliar state-action pairs to learn $P$ and $R$, and exploiting current knowledge about $P$ and $R$ to obtain more rewards. A classic solution to this problem is *UCB-VI* (Azar et al., 2017), which is used as the reference algorithm for STRIDE. To help the baselines work with long context history ($K \times H$ interactions in total), an external summarization of the past episodes is added in their prompts at the beginning of each new episode, similar to Krishnamurthy et al. (2024).

**Experiment Setup and Results.** In addition to STRIDE and the aforementioned baselines, we also include *UCB-VI* algorithm in the experiments, which serves as a reference. We evaluate on 10 randomly generated MDP instances with $H = 5$, $|\mathcal{S}| = 3$, and $|\mathcal{A}| = 3$, with the agents repetitively playing each instance for a total number of $K = 40$ episodes, and average the results over the 10 instances. In Figure 4, we report how the cumulative rewards collected in each episode change as the number of episodes experienced by the agent increases. STRIDE reliably implements the behavior of *UCB-VI* algorithm using the provided opera-

tions, and thus converges to the optimal policy at a similar rate as *UCB-VI*. In comparison, the baselines, though given additional summarization of history, fail to find the optimal policy as they cannot efficiently explore the environment.

### 3.2. Dynamic Mechanism Design

Section 3.1 presents the challenges of long-horizon planning and strategic exploration in MDP, which only involves a single agent. Here we further evaluate STRIDE (GPT-4o-2024-05-13 with temperature set to 0) on dynamic Vickrey-Clarke-Groves (VCG) mechanism design problem (Bergemann & Välimäki, 2019), a multi-agent generalization of MDP, which further necessitates the agent's ability to anticipate other agents' behaviors and plan accordingly.

**Agent's Objective in Dynamic Mechanism Design.** Consider a mechanism designer and a set of $N$ agents. The mechanism designer needs to elicit the reward functions $\{\widetilde{R}_i\}_{i=1}^N$ from the $N$ agents, with each $\widetilde{R}_i : \mathcal{S} \times \mathcal{A} \to \mathbb{R}$, and the agents can be untruthful. Based on reported reward functions, the designer chooses a policy $\pi : \mathcal{S} \to \Delta(\mathcal{A})$. At each step $h \in [H]$, the designer takes action $a_h \sim \pi(s_h)$, e,g., the allocation of some scarce resource among $I$ agents, and each agent $i \in [N]$ receives reward $R_i(s_h, a_h)$, i.e., agent $i$'s valuation for $a_h$ at state $s_h$. After $H$ steps of interactions, the designer needs to charge each agent $i$ some price $p_i \in \mathbb{R}$. The objective of each agent $i$ is to maximize its utility $u_i(\widetilde{R}_i) = V^\pi(P, R_i) - p_i$ by strategically report-

*Table 2.* Success rate in computing the VCG mechanism (10 runs).

| N | zero-shot CoT | zero-shot CoT w/ code | few-shot CoT w/ code | STRIDE |
|---|---|---|---|---|
| 2 | 0.69 | 0.63 | 0.70 | **0.89** |
| 4 | 0.57 | 0.63 | 0.54 | **0.90** |
| 6 | 0.49 | 0.45 | 0.44 | **0.86** |

ing the reward function $\widetilde{R}_i$. The objective of the designer is to maximize the expected cumulative sum of rewards, by strategically choosing the policy and pricing rule. This can be formulated as the following optimization problem

$$\pi^\star, \{p_i^\star\}_{i \in [N]} := \max_{\pi, \{p_i\}_{i \in [N]}} V^\pi(P, \sum_{i=1}^n \widetilde{R}_i) \quad (3)$$
$$s.t. \quad u_i(R_i) \geq u_i(R_i'), \forall R_i', i$$

where the constraint guarantees the incentive compatibility of all agents. The success rate for the experiments on this problem is computed by considering: (i) whether the chosen action $a_h$ satisfies $a_h = \pi_h^\star(s_h)$ for $h \in [H]$; and (ii) whether the charged price $p_i$ satisfies $|p_i - p_i^\star| \leq 0.01$.

**Experiment Setup and Results.** We evaluate on problem instances with horizon $H = 5$, number of states $|\mathcal{S}| = 3$, number of actions $|\mathcal{A}| = 3$, and number of agents $N \in \{2, 4, 6\}$. For each configuration, we repeat the experiment 10 times on randomly generated instances, by sampling dense tensors of size $\mathbb{R}^{S \times A \times S}$ and $\mathbb{R}^{N \times S \times A}$ as the transition function and reward functions for $N$ agents, respectively. The average success rate are reported in Table 2. We observe that the baselines, despite being capable of computing the optimal action most of the times, cannot generalize the same value iteration procedure to compute the VCG price correctly. In comparison, STRIDE can reliably compute the VCG price on most problem instances, which leads to its higher success rate.

### 3.3. Bargaining Games

We further evaluate STRIDE and the baselines (GPT-4o-2024-05-13 with the temperature set to 0) on bargaining games, in which a buyer and a seller engage in repeated negotiation for a finite number of steps. In order to maximize their utility, both the buyer and the seller need to predict the response of their opponent over long-horizon, based on the potentially incomplete information they have.

**Alternating Offer Bargaining under Complete Information.** We first consider the elementary yet seminal setting in which a buyer and a seller engage in a $T$-step bargaining process (with $T < \infty$) over price $p$ of the good. Specifically, at time step $t = 1$, the buyer offers a price to the seller and the game ends if the seller accepts the offer. Otherwise, the game continues to the next time step $t = 2$, where the seller makes a counteroffer. They repeat this process until the deadline $T$ is reached. Assuming the buyer's value for the

item is 1 and the seller's cost is 0, then the utility function of the buyer, denoted as $u_b$, and that of the seller, denoted as $u_s$, for some price $p$ at time step $t$ are

$$u_b(p, t) = (1 - p) \cdot \delta_b^{t-1}, \text{if } t \leq T, \text{and } 0 \text{ otherwise};$$
$$u_s(p, t) = (p - 0) \cdot \delta_s^{t-1}, \text{if } t \leq T, \text{and } 0 \text{ otherwise.} \quad (4)$$

respectively, with $\delta_b, \delta_s \in [0, 1]$ being the discount factor of their utilities over time. Note that in this setting, the buyer's value 1, the seller's cost 0, and the values of $\delta_b, \delta_s$ and $T$ are public information. The optimal decision for either agent, assuming his/her opponent is also acting optimally, i.e., being rational, is to play the Subgame Perfect Equilibrium (SPE) strategy, which, in this setting, is unique and can be computed using backward induction (Fudenberg & Tirole, 1991). Description of this reference algorithm and the operations constructed for STRIDE is given in Appendix B. To evaluate whether STRIDE and the baselines can make optimal decisions, we let buyer and seller empowered by the same method to bargain with each other, and report the success rates in reaching SPE.

**Experiment Setup and Results.** We evaluate on bargaining games with deadline $T \in \{3, 6, 9\}$. In each case, we repeat the experiments on 10 randomly generated instances, by sampling discount factors $\delta_b, \delta_s \in \mathcal{U}(0.5, 1.0)$. The average success rates are reported in Table 3. We can see that, none of the baseline methods attains success rate higher than 0.5, which is because when it is their turn to offer, they cannot offer a price close to SPE, though being explicitly instructed in the prompt to assume rational opponent behavior when making decisions. It is worth noting that the existence of the code interpreter did not provide any advantage this time compared with the results for MDP. Though the LLM did attempt to implement the backward induction algorithms to solve SPE, they failed to get everything right and produce the correct results. We hypothesize that this distinction is due to the insufficiency of training data related to the implementation of backward induction algorithms for bargaining, especially compared with the algorithms for MDP.

Moreover, to further illustrate the advantage of being able to strategically reason about the decisions in bargaining, we pit STRIDE against zero-shot CoT, the best-performing baseline in Table 3. The results (averaged over 10 randomly generated instances) are summarized in Table 4. We can see that, by mimicking the reference algorithm, STRIDE guarantees an outcome that is no worse than the SPE regardless

*Table 3.* Success rate in reaching SPE of single-issue bargaining (10 runs).

| $T$ | zero-shot CoT | zero-shot CoT w/ code | few-shot CoT w/ code | STRIDE |
|---|---|---|---|---|
| 3 | 0.50 | 0.35 | 0.50 | **0.79** |
| 6 | 0.50 | 0.27 | 0.46 | **0.91** |
| 9 | 0.34 | 0.18 | 0.27 | **0.74** |

*Table 4.* Outcomes of STRIDE and zero-shot CoT bargaining with each other.

| | STRIDE buyer vs zero-shot CoT seller | | zero-shot CoT buyer vs STRIDE seller | |
|---|---|---|---|---|
| $T$ | avg SPE price | avg sale price | avg SPE price | avg sale price |
| 3 | 0.13 | 0.13 | 0.22 | 0.43 |
| 6 | 0.57 | 0.56 | 0.65 | 0.70 |
| 9 | 0.28 | 0.27 | 0.49 | 0.70 |

*Table 5.* Success rate in reaching SE of single-issue bargaining with one-sided uncertainty (10 runs).

| $T$ | zero-shot CoT | zero-shot CoT w/ code | few-shot CoT w/ code | STRIDE |
|---|---|---|---|---|
| 3 | 0.47 | 0.29 | 0.38 | **0.79** |
| 6 | 0.44 | 0.32 | 0.30 | **0.75** |
| 9 | 0.49 | 0.38 | 0.23 | **0.69** |

of the role it plays. As mentioned in the previous paragraph, the baseline cannot accurately compute the SPE price, and thus, when it serves as the buyer who needs to make the initial offer, often ends up with a sale price that is higher than SPE price, which demonstrates its sub-optimality.

**Seller Making Offers under Uncertainty of Buyer's Value.** Now we consider the more challenging scenario where the buyer's value, denoted as $b \in [0, 1]$, is privately known to himself, and thus the seller needs to update his/her belief about $b$ based on the observed responses, i.e., buyer's rejection of seller's offers. The seller's cost (still assumed to be 0) and the prior distribution of $b$, represented as a cumulative distribution function $F(v)$, are public information. $F(\cdot)$ is supported on $[0, 1]$ and we assume $F(v) = v$, i.e., a uniform distribution. In each time step $t = 1, 2, \ldots, T$, the seller offers a price and the buyer responds by acceptance or rejection. Similar to Eq (4), their utility functions are

$$u_b(p, t) = (b - p) \cdot \delta_b^{t-1}, \text{if } t \leq T, \text{and } 0 \text{ otherwise,}$$
$$u_s(p, t) = p \cdot \delta_s^{t-1}, \text{if } t \leq T, \text{and } 0 \text{ otherwise.} \tag{5}$$

Different from the complete information setting where we evaluate the agents using the unique SPE, here we consider sequential equilibrium (SE) due to the uncertainty on the buyer's value. Fortunately, in the particular setting described above, the SE is still unique (Cramton, 1984), and thus we can similarly evaluate the agents using the success rates of reaching SE. To compute the SE, we propose a reference algorithm for STRIDE that combines bisection search and backward induction and construct the specialized tools. More details are given in Appendix B.

**Experiment Setup and Results.** We evaluate the agents on problems with deadline $T \in \{3, 6, 9\}$. In each case, we re-

peat the experiments on 10 randomly generated instances, by sampling discount factors $\delta_b, \delta_s \in \mathcal{U}(0.5, 1.0)$ and buyer's value $b \in \mathcal{U}(0.1, 0.9)$. The average success rates are reported in Table 5. Again, we observe that STRIDE outperforms the baseline methods, as it is able to compute the SE by mimicking the reference algorithm we designed.

## Conclusion

This paper presented the STRIDE framework, enhancing LLMs' strategic decision-making capabilities. Through integrating a structured *Thought* process with external working memory and operational tools, STRIDE enables LLMs to overcome significant limitations such as strategic exploration and dynamic opponent interaction. Our evaluations across diverse decision-making scenarios validate STRIDE's effectiveness, suggesting its potential as a robust tool for strategic thinking in complex environments. For further development of the STRIDE framework, we propose the following research avenues. (i) Currently, STRIDE utilize specially designed Python functions as tools to facilitate the formation of strategies and the choice of actions by the agents in bilateral bargaining, an interesting direction is to replace it with models trained using data collected during interactions. (ii) Automatic synthesis of operations: Another interesting direction would be developing LLMs specifically fine-tuned on implementing primitives that handle the low-level calculations of various decision-making problems. (iii) Fine-tuning on the *Thought* Sequence: To further enhance LLM's understanding and execution of the *Thought* sequence as well as the associated operations, we can fine-tune the model on the sequences generated using the procedure described in Section 2.2.

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

# A. Related Work

**Evaluating LLMs' Reasoning Capability in Strategic Environments.** Recent advancements in LLMs have spurred investigations into their capacity for strategic reasoning. There have been several contributions recently studying the behavior of LLMs in strategic settings, e.g., matrix games like Dictator and Prisoner's Dilemma (Brookins & DeBacker, 2023; Lorè & Heydari, 2023; Fan et al., 2023; Akata et al., 2023; Guo et al., 2023). These works have been particularly interested in assessing whether LLMs can perform strategic or rational reasoning effectively with minimal initial input, often referred to as zero-shot prompting. Recent work by Davidson et al. (2024) and Bianchi et al. (2024) also used bargaining games to evaluate the strategic reasoning of LLMs. Their findings in general suggest that while LLMs can sometimes generate plausible strategies, they often lack consistency and a deep understanding of game dynamics. Another pivotal study by Gandhi et al. (2023) proposed to enhance the strategic reasoning of LLMs by providing few-shot chain-of-thought (CoT) examples for matrix games and multi-turn bargaining games, and showed that LLMs are capable of generalizing the demonstration to new game instances, but still has difficulties in games with complex rules or long horizon. A recent work by Huang et al. (2024) also made a similar observation about the limited capability of LLMs to generalize to various game instances, despite using CoT to enhance reasoning.

**Applications of LLM-based Agents beyond Strategic Reasoning.** While our focus is on enhancing the strategic reasoning capabilities of LLMs, it is important to acknowledge the broader applications of LLM-based agents that do not primarily focus on strategic tasks (Wang et al., 2024), such as social simulation, e.g., building virtual environment with LLM-based agents to simulate social phenomena (Park et al., 2023; Aher et al., 2023), scientific research assistant, e.g., utilize LLMs for automating the design, planning, and execution of scientific experiments (Boiko et al., 2023), software development, e.g., let multiple LLM agents communicate and collaborate through natural language to complete the software development life cycle (Qian et al., 2023), and robotics, e.g., equip LLM with a wide range of manipulation and navigation skills to control a mobile manipulator robot (Ahn et al., 2022). This wide range of applications has led to various design of the LLM agent architecture to enhance its capability in the specific domain, but typically, an LLM agent architecture features memory (Zhu et al., 2023) and planning (Yao et al., 2022) modules that enable LLMs to recall past behaviors and plan future actions in a dynamic environment, and a set of tools (Qin et al., 2023) that facilitate mathematical computation, accessing internal or external memory, and interacting with the environment.

**LLM-Enhanced Reinforcement Learning Algorithms.** The works mentioned in the previous two paragraphs, as well as the STRIDE framework proposed in this paper, utilize LLMs as the decision maker, that is, LLMs are fed prompts containing the current state of the environment, and they generate action recommendations based on this input. The reasoning process that produces the recommendation, regardless of whether it follows certain algorithmic behavior as STRIDE, happens in the language space. Another distinct line of research, emerging primarily from the reinforcement learning community, instead integrates LLMs into traditional reinforcement learning algorithms to leverage the common sense knowledge that LLMs acquire during pretraining (Hao et al., 2023; Liu et al., 2023; Zhou et al., 2023; Zhao et al., 2024). In this way, the reasoning process is hard-coded in programming language like Python, which defines how different components interact with each other. Currently, the most prevalent approach in this domain is the integration of LLMs into Monte Carlo tree search (MCTS) algorithms, where they typically serve as tree traversal policy (Zhao et al., 2024), action pruner (Liu et al., 2023), world model (Hao et al., 2023), and evaluation function (Liu et al., 2023). In comparison, our approach is much more flexible in the sense that we can repurpose the reasoning process of STRIDE to emulate different algorithmic behaviors using various tools and demonstrations. In particular, as demonstrated in our additional experiments, apart from the model-based RL algorithms like UCB-VI, we can also make STRIDE reason as tree-search algorithms like Minimax. And as discussed in Section 2.2, this flexibility extends the utility of our approach well beyond decision-making problems.

# B. Reference Algorithms for STRIDE

As discussed in Section 2, the main strength of STRIDE lies in its capability of implementing various algorithmic behaviors in its *Thought* process to solve decision-making problems that are challenging to LLMs. In this section, we provide the descriptions of the reference algorithms that STRIDE emulates when solving the problems in Section 3.

## B.1. Value Iteration & Upper Confidence Bound Value Iteration

For MDP with known and unknown model, the reference algorithms selected for STRIDE are Value Iteration (VI) and Upper Confidence Bound Value Iteration (UCB-VI). Here we provide description of these two algorithms in Algorithm 1 and Algorithm 2, as well as some simplified comments (e.g., results returned by the operations are omitted for simplicity)

---

**Algorithm 1** Value Iteration for MDPs with Known Model.

---

1: **Initialize** $V_{H+1}(s) = 0, \forall s \in \mathcal{S}$
2: ▷ **Question: Compute the Optimal Policy.**
3: **for** step $h = H, H - 1, \cdots, 1$ **do**
4:     ▷ Thought: Now we can continue to compute the Q-values for the current step $h$.
5:     ▷ Operation: call `UpdateQbyR` with inputs {time_step: h}
6:     ▷ Operation: call `UpdateQbyPV` with inputs {time_step: h}
7:     ▷ Operation: call `UpdateVbyQ` with inputs {time_step: h}
8:     **for** each state $s \in \mathcal{S}$ **do**
9:         **for** each action $a \in \mathcal{A}$ **do**
10:            $Q_h(s, a) = R(s, a) + \sum_{s' \in \mathcal{S}} P(s'|s, a) V_{h+1}(s')$
11:        $V_h(s) = \max_{a \in \mathcal{A}} Q_h(s, a)$
12: ▷ Thought: I have finished value iteration. Now exit reasoning.
13: **for** step $h = 1, 2, \cdots, H$ **do**
14:     Observe state $s_h$
15:     ▷ **Question: Which action I should take?**
16:     ▷ Thought: I should choose the action that maximizes the computed Q values.
17:     ▷ Operation: call `GetQ` with inputs {time_step: h, cur_state: $s_h$}
18:     ▷ Operation: call `GetArgMax` with inputs {q_vals: [. . . ]}
19:     ▷ Exit: I should choose Action $a_h$ as it maximizes the Q values. Now exit reasoning.
20:     Take action $a_h = \arg\max_{a \in \mathcal{A}} Q_h(s_h, a)$
21:     Observe reward $r(s_h, a_h) = R(s_h, a_h) + \epsilon$ and state transits to $s_{h+1}$

---

showing how we augment the algorithm during the demonstration generation procedure as discussed in Section 2.2.

**Operational Tools.** The following operational tools are provided to the LLM to help it implement the behavior of VI and UCB-VI:

- `UpdateQbyR`: add reward $R(s, a)$ to $Q_h(s, a)$ for all $(s, a)$ pairs at the specified time step $h$,
- `UpdateQbyPV`: add one-step look-ahead value $\sum_{s' \in \mathcal{S}} P(s'|s, a) V_{h+1}(s')$ to $Q_h(s, a)$ for all $(s, a)$ pairs at the specified time step $h$,
- `UpdateV`: take maximum $V_h(s) = \max_{a \in \mathcal{A}} Q_h(s, a)$ for the specified time step $h$,
- `GetQ`: retrieve the values of $Q_h(s, a)$ for all action $a \in \mathcal{A}$ at the specified time step $h$ and state $s$,
- `GetArgMax`: return the indices corresponding to the maximal value in the given list of numbers
- `UpdateQbyBonus`: add exploration bonus to the Q values for all state-action pairs at the specified time step
- `UpdateMDPModel`: update the estimation of the reward and transition function of MDP based on the observed quadruple (old state, action, new state, reward)

**MDP with Known Model.** With these operational tools, STRIDE is capable of computing the optimal policy of MDP with known model by emulating Algorithm 1.

**MDP with Unknown Model.** Similarly, STRIDE can emulate Algorithm 2 when facing MDP with unknown model, which only needs two additional operations that (i) update the estimation for the unknown reward and transition function, and (ii) update Q values with the exploration bonus, respectively.

### B.2. Dynamic Programming for Dynamic Mechanism Design

For dynamic mechanism design problem, the reference algorithm selected for STRIDE is described in Algorithm 3, which is modified based on the Markov VCG mechanism of Lyu et al. (2022). It is known that the unique solution to Eq (3) is the VCG mechanism i.e.,

$$\pi^\star := \arg\max_\pi V^\pi(P, \textstyle\sum_{i=1}^N \widetilde{R}_i),$$
$$p_i^\star := V^{\pi^*_{-i}}(P, \textstyle\sum_{j \neq i} \widetilde{R}_j) - V^{\pi^*}(P, \textstyle\sum_{j \neq i} \widetilde{R}_j), \quad \text{for } i = 1, 2, \ldots, n,$$

---

**Algorithm 2** Value Iteration Upper Confidence Bound for MDPs with Unknown Model

---

1: **Initialize** $V_{H+1}(s) = 0, \forall s \in \mathcal{S}$
2: **for** episode $t = 1, 2, \ldots, T$ **do**
3:    ▷ **Question: Compute the Optimistic Policy for Exploration.**
4:    **for** step $h = H, H-1, \cdots, 1$ **do**
5:       ▷ Thought: Now we can continue to compute the Q-values for the current step $h$.
6:       ▷ Operation: call UpdateQbyR with inputs {time_step: h}
7:       ▷ Operation: call UpdateQbyPV with inputs {time_step: h}
8:       ▷ Operation: call UpdateQbyBonus with inputs {time_step: h}
9:       ▷ Operation: call UpdateVbyQ with inputs {time_step: h}
10:      **for** each state $s \in \mathcal{S}$ **do**
11:         **for** each action $a \in \mathcal{A}$ **do**
12:            ▷ Action: call Python function to calculate Q value for $(s, a)$
13:            $Q_h(s, a) = \widehat{R}(s, a) + \sum_{s' \in \mathcal{S}} \widehat{P}(s'|s, a) V_{h+1}(s') + b(N(s, a))$
14:         $V_h(s) = \max_{a \in \mathcal{A}} Q_h(s, a)$
15:   ▷ Thought: I have finished value iteration. Now exit reasoning.
16:   **for** step $h = 1, 2, \cdots, H$ **do**
17:      Observe state $s_h$
18:      ▷ **Question: Which action I should take?**
19:      ▷ Thought: I should choose the action that maximizes the computed Q values.
20:      ▷ Operation: call GetQ with inputs {time_step: h, cur_state: $s_h$}
21:      ▷ Operation: call GetArgMax with inputs {q_vals: [. . .]}
22:      ▷ Exit: I should choose Action $a_h$ as it maximizes the Q values. Now exit reasoning.
23:      Take action $a_h = \arg\max_{a \in \mathcal{A}} Q_h(s_h, a)$
24:      Observe reward $r(s_h, a_h) = R(s_h, a_h) + \epsilon$ and state transits to $s_{h+1}$
25:      ▷ **Question: Update estimations of $P$ and $R$.**
26:      ▷ Thought: I should update my estimation using the observed $(s_h, a_h, s_{h+1}, r_h)$.
27:      ▷ Operation: call UpdateMDPModel with inputs {s: $s_h$, a: $a_h$, s_prime: $s_{h+1}$, r: $r_h$}
28:      ▷ Thought: My estimation is updated. Now exit reasoning.
29:      $N(s_h, a_h) = N(s_h, a_h) + 1$, $N(s_h, a_h, s_{h+1}) = N(s_h, a_h, s_{h+1}) + 1$
30:      $\widehat{P}(s_{h+1}|s_h, a_h) = \frac{N(s_h, a_h, s_{h+1})}{N(s_h, a_h)}$, $\widehat{R}(s, a) = \widehat{R}(s, a) \times \frac{N(s_h, a_h) - 1}{N(s_h, a_h)} + \frac{r(s_h, a_h)}{N(s_h, a_h)}$

---

where $\pi_{-i}^* := \arg\max_\pi V^\pi(P, \sum_{j \neq i} \widetilde{R}_j)$. Similar to Eq (1), Eq (B.2) can be solved by separately computing policies $\pi^\star$ and $\{\pi_{-i}^*\}_{i=1}^N$ via value iteration, and then evaluating $\pi^\star$ on MDP instances with transition function $P$ and reward function $\sum_{j \neq i} \widetilde{R}_j$ for $i = 1, 2, \ldots, N$.

**Operational Tools.** The following operational tools are provided to the LLM:

- UpdateQbyRExcluding: add immediate rewards, excluding the reward of excluded_agent, to the Q values for all state-action pairs at current time step. If excluded_agent is set to None, all agents' rewards are used.
- UpdateQbyPVExcluding: add the one-step look-ahead value, excluding the reward of excluded_agent, to the Q values for all state-action pairs at current time step. If excluded_agent is set to None, all agents' rewards are used.
- UpdateVExcluding: update the V values, excluding the reward of excluded_agent, based on the computed Q values for the current time step. If excluded_agent is set to None, all agents's rewards are used.
- GetQExcluding: retrieve Q values, that excludes the rewards of excluded_agent, for all actions at the current state and time step. If excluded_agent is set to None, the Q values computed using all agents' rewards will be returned.
- EvaluatePolicyExcluding: evaluate the optimal policy on an fictitious MDP that excludes the reward function of excluded_agent.
- GetArgMax: return the indices corresponding to the maximal value in the given list of numbers
- GetMax: return the maximal value in the given list of numbers

With these operational tools, STRIDE is capable of computing the dynamic VCG mechanism by emulating Algorithm 3.

---

**Algorithm 3** Dynamic VCG Mechanism Design

---

1: **Initialize** $V_{H+1}(s) = 0, V_{H+1,-i}(s) = 0, \forall s \in \mathcal{S}$
2: ▷ **Question: Compute the optimal policy that maximizes all agents' reported rewards.**
3: **for** step $h = H, H-1, \cdots, 1$ **do**
4:    ▷ Thought: Now we can continue to compute the Q-values for the current step $h$.
5:    ▷ Operation: call `UpdateQbyRExcluding` with {time_step: h, excluded_agent:None}
6:    ▷ Operation: call `UpdateQbyPVExcluding` with {time_step: h, excluded_agent:None}
7:    ▷ Operation: call `UpdateVbyQExcluding` with {time_step: h, excluded_agent:None}
8:    **for** each action $a \in \mathcal{A}$ **do**
9:       $Q_h(s,a) = \sum_i^N R_i(s,a) + \sum_{s' \in \mathcal{S}} P(s'|s,a)V_{h+1}(s')$
10:    $V_h(s) = \max_{a \in \mathcal{A}} Q_h(s,a)$
11: ▷ Thought: I have finished value iteration. Now exit reasoning.
12: Denote the optimal policy as $\pi_h^\star(s) := \arg\max_{a \in \mathcal{A}} Q_h(s,a)$ for $h \in [H], s \in \mathcal{S}$
13: **for** step $h = 1, 2, \cdots, H$ **do**
14:    Observe state $s_h$
15:    ▷ **Question: Which action I should take?**
16:    ▷ Thought: I should choose the action that maximizes the computed Q values.
17:    ▷ Operation: call `GetQExcluding` with {time_step: h, cur_state: $s_h$, excluded_agent=None}
18:    ▷ Operation: call `GetArgMax` with {q_vals: [. . .]}
19:    ▷ Exit: I should choose Action $a_h$ as it maximizes the Q values. Now exit reasoning.
20:    Mechanism designer takes action $a_h = \arg\max_{a \in \mathcal{A}} Q_h(s_h, a)$
21:    Agent $i$ observes reward $r_i(s_h, a_h) = R_i(s_h, a_h) + \epsilon$ for $i \in [N]$ and state transits to $s_{h+1}$
22: **for** agent $i = 1, 2, \cdots, N$ **do**
23:    ▷ **Question: Now compute the VCG price for agent $i$.**
24:    **for** step $h = H, H-1, \cdots, 1$ **do**
25:       ▷ Thought: Now we can continue to compute the Q-values for the current step $h$.
26:       ▷ Operation: call `UpdateQbyRExcluding` with {time_step: h, excluded_agent: $i$}
27:       ▷ Operation: call `UpdateQbyPVExcluding` with {time_step: h, excluded_agent: $i$}
28:       ▷ Operation: call `UpdateVbyQExcluding` with {time_step: h, excluded_agent: $i$}
29:       **for** each state $s \in \mathcal{S}$ **do**
30:          **for** each action $a \in \mathcal{A}$ **do**
31:             $Q_{h,-i}(s,a) = \sum_{j \neq i} R_j(s,a) + \sum_{s' \in \mathcal{S}} P(s'|s,a)V_{h+1,-i}(s')$
32:          $V_{h,-i}(s) = \max_{a \in \mathcal{A}} Q_{h,-i}(s,a)$
33:    $p_i^\star = V_{1,-i}(s_1) - V^{\pi^*}(P, \sum_{j \neq i} \widetilde{R}_j)$
34:    ▷ Thought: Now we know the optimal value of this fictitious MDP that ignores agent $i$'s rewards. Next we should evaluate policy $\pi^\star$ on this fictitious MDP.
35:    ▷ Operation: call `EvaluatePolicyExcluding` with {excluded_agent: $i$}
36:    ▷ Thought: Then the VCG price for agent $i$ is simply their difference ... Now exit reasoning.

---

### B.3. Backward Induction for Bargaining in Complete Information Setting

For alternating offer bargaining under complete information, the reference algorithm selected for STRIDE is the backward induction algorithm described in Algorithm 4, which given parameter of the game, including buyer's discount $\delta_b$, seller's discount $\delta_s$, and deadline $T$, can compute the SPE of the game.

**Operational Tools.** The following operational tools are provided to the LLM:

- `CalcUtil`: calculate buyer or seller's utility using Eq (4), with the role of the agent, the specified price and time step as inputs.
- `BackwardOneStep`: compute the SPE price using one step of backward induction reasoning based on the opponent's utility if he/she choose to reject the offer at current time step (see the constrained optimization problem in line 14 and line 17 in Algorithm 4)
- `GetSPEPrice`: retrieve the previously computed SPE price for the specified time step

---

**Algorithm 4** Backward Induction to Compute SPE of Bargaining under Complete Information

---

1: ▷ **Question: Compute the SPE Prices via Backward Induction.**
2: **for** time step $t = T, T-1, \cdots, 1$ **do**
3:     ▷ Thought: Compute the SPE price for time $t$, based on the results computed for time $t+1$
4:     **if** $t = T$ **then**
5:         **if** current_player = Buyer **then**
6:             ▷ Operation: call BackwardOneStep with {agent: buyer, op_u: 0.0, t: $T$}
7:             The SPE price $p_T := 0.0$
8:         **else**
9:             ▷ Operation: call BackwardOneStep with {agent: seller, op_u: 0.0, t: $T$}
10:             The SPE price $p_T := 1.0$
11:     **else**
12:         **if** current_player = Buyer **then**
13:             ▷ Operation: call BackwardOneStep with {agent: buyer, op_u: $u_s(p_{t+1}, t+1)$, t: $t$}
14:             The SPE price $p_t := \arg\max_p u_b(p, t)$, s.t. $u_s(p, t) \geq u_s(p_{t+1}, t+1)$
15:         **else**
16:             ▷ Operation: call BackwardOneStep with {agent: seller, op_u: $u_b(p_{t+1}, t+1)$, t: $t$}
17:             The SPE price $p_t := \arg\max_p u_s(p, t)$, s.t. $u_b(p, t) \geq u_b(p_{t+1}, t+1)$.
18:     ▷ Operation: call CalcUtil with {agent: seller, price: $p_t$, t: $t$}
19:     ▷ Operation: call CalcUtil with {agent: buyer, price: $p_t$, t: $t$}
20:     Buyer utility $u_b(p_t, t)$, Seller utility $u_s(p_t, t)$
21: ▷ Thought: SPE prices for all time steps are calculated. Now exit reasoning.

---

With these operational tools, STRIDE is capable of computing the SPE by emulating Algorithm 4. SPE can be used to predict the future offer to be made by the opponent, assuming the opponent is rational and that the opponent believes the player to be rational as well. When facing a new offer $p$ made by the opponent at time step $t$, STRIDE will emulate Algorithm 5 to produce a response.

---

**Algorithm 5** Response to Offer in Bargaining with Complete Information

---

1: **Inputs:** current_player, price $p$, time $t$, SPE prices $\{p_t\}_{t=1}^T$
2: ▷ **Question: Should I accept or reject opponent's offer?**
3: ▷ Thought: I should first compute the utility I get by accepting the offer, and then the utility I get by rejecting the offer and making a counter offer using the SPE price in the next time step.
4: ▷ Operation: call CalcUtil with inputs {agent: current_player, price: $p$, t: $t$}
5: ▷ Operation: call GetSPEPrice with inputs {t: $t+1$}
6: ▷ Operation: call CalcUtil with inputs {agent: current_player, price: $p_{t+1}$, t: $t+1$}
7: **if** current_player = buyer **then**
8:     $u_a = u_b(p, t)$, $u_r = u_b(p_{t+1}, t+1)$
9: **else**
10:     $u_a = u_s(p, t)$, $u_r = u_s(p_{t+1}, t+1)$
11: **if** $u_a \geq u_r$ **then**
12:     ▷ Thought: I should accept the offer. Now exit reasoning.
13:     return Accept
14: **else**
15:     ▷ Thought: I should reject the offer. Now exit reasoning.
16:     return Reject

---

### B.4. Backward Induction for Bargaining in Incomplete Information Setting

Since the seller is uncertain about the value $b$ of the buyer, at each time step $t$ the seller decides the offer price $p_t$ based on his/her belief constructed using observations up to time step $t-1$, which is denoted as $\mathcal{U}(0, b_{t-1})$, i.e., the true value $b$ is

---

**Algorithm 6** Backward Induction to Compute SE of Bargaining under Incomplete Information

---

1: ▷ **Question: Compute the SE Prices via Bisection Search and Backward Induction.**

2: ▷ Thought: I need to first compute my belief about buyer's value at time step T-1 under sequential equilibrium, denoted $b_{T-1}$, which can be done via bisection search. I should terminate when the value $b'_0$ computed based on $b'_{T-1}$ gets close enough to my actual initial belief $b_0 = 1.0$.

3: $l = 0, h = 1, B'_{T-1} = (l+h)/2$

4: ▷ Operation: Call `ComputeBt` with inputs {time_step: 1, b_last: $B'_{T-1}$}

5: $b'_0 = \texttt{ComputeBt}(1, b'_{T-1})$

6: **while** $|b'_0 - 1.0| \geq 10^{-3}$ **do**

7:     **if** $b'_0 \leq 1.0$ **then**

8:         ▷ Thought: Since $b'_0$ is smaller than $b_0$, I should focus on the region $[b'_{T-1}, h]$ next time.

9:         $l = b'_{T-1}$

10:     **else**

11:         ▷ Thought: Since $b'_0$ is larger than $b_0$, I should focus on the region $[l, b'_{T-1}]$ next time.

12:         $h = b'_{T-1}$

13:     $b'_{T-1} = (l+h)/2$

14:     ▷ Operation: Call `ComputeBt` with inputs {time_step: 1, b_last: $B'_{T-1}$}

15:     $b'_0 = \texttt{ComputeBt}(1, b'_{T-1})$

16: ▷ Thought: Since $|b'_0 - 1.0| < 10^{-3}$, the value of my initial belief computed based on $B'_{T-1}$ is close enough to the actual value $b_0 = 1$. Therefore, $B'_{T-1}$ is an accurate approximation of $B_{T-1}$ in SE. Now I can start backward induction to compute the SE prices from time $T$ to 1.

17: **for** $t = T, T-1, \ldots, 1$ **do**

18:     **if** $t = T$ **then**

19:         ▷ Operation: Call function `SOLVELAST` with inputs {b_last: $B'_{T-1}$}.

20:         $u_t, p_t = \texttt{SolveLast}(B'_{T-1})$ # seller's expected utility and price under SE

21:     **else**

22:         ▷ Operation: Call function `SOLVE` with inputs {u: $u_{t+1}$, p: $p_{t+1}$, t: $t$}.

23:         $u_t, p_t = \texttt{Solve}(u_{t+1}, p_{t+1}, t)$ # seller's expected utility and price under SE

24:     ▷ Thought: Now I need to continue to time step $t-1$.

25: ▷ Thought: I have reached $t = 1$. Exit reasoning now.

---

uniformly distributed in $[0, b_{t-1}]$ (with $b_0 = 1$). Therefore, different from SPE considered in complete information setting, SE specifies not only the strategies of the players, but also the belief, which in our case is the sequence $\{b_0, b_1, \ldots, b_{T-1}\}$. In classic economics literature (Sobel & Takahashi, 1983; Cramton, 1984), this sequence is obtained by: (i) backward induction from time $T$ to time 1, which results in $b_0$ expressed as a function of $b_{T-1}$; (ii) as the initial belief $b_0 = 1$, we can solve this equation to obtain the value of $b_{T-1}$. This provides an analytical form for $\{b_0, b_1, \ldots, b_{T-1}\}$ using the parameters $\delta_b, \delta_s, T$. To make the inner logic more transparent during reasoning, we replace this analytical solution with a bisection search when designing the reference algorithm for STRIDE, with its full description given in Algorithm 6.

We provide the following operational tools to STRIDE to help it emulate Algorithm 6:

- `CalcUtil`: calculate buyer or seller's utility using Eq (5), with the role of the agent, the specified price and time step as inputs.
- `ComputeBt`: compute what seller's belief about buyer's value would be at the current time step, given a guess of seller's belief at time step $T-1$ (description given in Algorithm 7)
- `SolveLast`: compute seller's expected utility and the corresponding price at the last time step (description given in Algorithm 8)
- `Solve`: compute the expected utility and the corresponding price at the current time step, based on the results computed for the next time step (description given in Algorithm 9)
- `GetSEPrice`: retrieve the previously computed SE price for the specified time step

Then similar to the complete information setting, when deciding whether to accept an offer from the seller, the buyer can compare the utility he/she can get by accepting the current offer, and the utility he/she can get by waiting for seller's offer in

the next time step. For the latter, as the buyer assumes the seller is rational, the next offer from seller is predicted using the SE price from Algorithm 6.

---

**Algorithm 7** `ComputeBt`

---

1: **Inputs** time_step, the time index of current belief, and b_last, the belief at time step $T$.
2: **Initialize** constants $\{c_\tau\}_{\tau=2}^{T}$ with $c_T = 0.5$ and $c_\tau = \frac{(1-\delta_b+\delta_b c_{\tau+1})^2}{2(1-\delta_b+\delta_b c_{\tau+1})-\delta_s c_{\tau+1}}$ for $\tau \geq 2$.
3: Set $t = $ time_step, $b_{T-1} = $ b_last
4: **for** $\tau = T-1, T-2, \ldots, t$ **do**
5: $\qquad b_{\tau-1} = \frac{2(1-\delta_b+\delta_b c_{\tau+1})-\delta_s c_{\tau+1}}{1-\delta_b+\delta_b c_{\tau+1}} b_\tau$
6: **return** $b_{t-1}$

---

**Algorithm 8** `SolveLast`

---

1: **Inputs** b_last, the belief at time step $T$.
2: Set $b_{T-1} = $ b_last
3: Compute SPE price $p_T := \arg\max_p p \cdot \frac{b_{T-1}-p}{b_{T-1}} = \frac{1}{2}b_{T-1}$
4: Compute expected utility $u_T := p_T \cdot \frac{b_{T-1}-p_T}{b_{T-1}} = \frac{1}{4}b_{T-1}$
5: **return** $u_T, p_T$

---

**Algorithm 9** `Solve`

---

1: **Inputs** u, seller's expected utility at t+1, p, the associated price, and t, the current time step.
2: Set $u_{t+1} = $ u, $p_{t+1} = $ p, and $t = $ t
3: Compute SPE price

$$p_t := \arg\max_p \frac{b_{t-1}-b_t}{b_{t-1}}p + \frac{b_t}{b_{t-1}}u_{t+1}, \text{ s.t. } b_t = \delta_b(b_t - p_{t+1})$$
$$= (1-\delta_b)b_t + \delta_b p_{t+1}$$

4: Compute expected utility $u_t = \frac{b_{t-1}-b_t}{b_{t-1}}p_t + \frac{b_t}{b_{t-1}}u_{t+1}$
5: **return** $u_t, p_t$

---

## C. Prompts of the STRIDE Framework and Baselines

The prompts used for the LLM agents in Section 3 consist of three parts, which we mark using different colors in this section: a system prompt setting the role of the agent (gray), followed by a formal description of the decision-making problem to be solved (light blue), and then parameters of the problem instance (light green). The system prompt is problem-agnostic, which is given below.

---

**System prompt for zero-shot CoT**

You are a world class intelligent agent capable of solving various classes of decision making problems. For each decision making problem you encounter next, you will be given the description of the problem setup and your objective. You need to carefully reason about the problem step-by-step, and make optimal decisions for the encountered problem instance.

---

**System prompt for zero-shot CoT w/ code interpreter**

You are a world class intelligent agent capable of solving various classes of decision making problems. For each

---

decision making problem you encounter next, you will be given the description of the problem setup and your objective. You need to carefully reason about the problem step-by-step, and make optimal decisions for the encountered problem instance. You are provided with a code interpreter. You should write and run code to answer the questions.

---

**System prompt for few-shot CoT w/ code interpreter**

You are a world class intelligent agent capable of solving various classes of decision making problems. For each decision making problem you encounter next, you will be given the description of the problem setup and your objective. Your need to carefully reason about the problem, and make optimal decisions for the encountered problem instance. You are provided with a code interpreter and an example implementation. You should write and run code to answer the questions following the example.

---

**System prompt for STRIDE**

You are a world class intelligent agent capable of solving various classes of decision making problems. For each decision making problem you encounter next, you will be given the description of the problem setup and your objective. Your need to carefully reason about the problem step-by-step, and make optimal decisions for the encountered problem instance. You are provided with a set of tools that handle low-level calculations and examples showing you how to use these tools to solve this problem.

---

In the remainder of this section, we will provide the prompts describing the decision making problems and the problem parameters to the agents.

**C.1. MDP with Known Model**

The following are the prompts we provide to all agents to describe the formulation and the agent's objective in MDP when the model, i.e., the transition function and reward function, is known.

---

**Description of MDP with known model**

A finite horizon tabular Markov Decision Process (MDP) is a model for decision-making in scenarios where outcomes are influenced by both randomness and controlled decisions, with decisions being made over a finite number of time steps.

Components:

State Space $S$: $s_0, s_1, \ldots, s_{|S|-1}$, where $|S|$ is the total number of states.

Action Space $A$: $a_0, a_1, \ldots, a_{|A|-1}$, where $|A|$ is the total number of actions.

Transition probability matrix $P$: a three-dimensional tensor with shape $|S| \times |A| \times |A|$, where each entry represents the probability of transitioning from one state after taking a specific action to another state.

Reward matrix $R$: a matrix with shape $|S| \times |A|$, where each entry gives the mean of the immediate reward received after taking an action in a state.

Horizon length $H$: The total number of time steps the decision process is constrained to.

Interaction protocol:

For time step $h = 1, 2, \ldots, H$

Agent takes an action $a_h \in A$ based on the current state $s_h$

---

Agent receives reward $r_h := R[s_h, a_h] + \eta_h$, where $\eta_h \sim \mathcal{N}(0, 1)$

The environment transits to the next state $s_{h+1}$ with probability $P[s_h, a_h, s_{h+1}]$ Goal of the agent:

Maximize expected cumulative rewards $\mathbb{E}\left[\sum_{h=1}^{H} R[s_h, a_h]\right]$, where the expectation is w.r.t. randomness of agent's policy and state transition.

For zero-shot CoT, which can only read the parameters from context, we print the complete transition matrix $P$ and reward matrix $R$ as shown below, where the empty curly brackets {} are substituted with actual values of the problem instance.

---

**Description of problem instance**

Now you are going to play in a finite-horizon tabular Markov decision process, with length of horizon {} (with time indices starting from h=0 to {}), number of states —S—={}, number of actions —A—={}. The transition matrix P is: {} and reward matrix R is {}.

---

For zero-shot CoT w/ code, few-shot CoT w/ code and STRIDE, which can read the parameters from their working memory or an external file, instead of directly printing the transition and reward matrices in context, we state in the prompt where these values can be accessed.

---

**Description of problem instance**

Now you are going to play in a finite-horizon tabular Markov decision process, with length of horizon {} (with time indices starting from h=0 to {}), number of states —S—={}, number of actions —A—={}. The transition matrix P and reward matrix R are stored in working memory.

---

### C.2. MDP with Unknown Model

The following are the prompts we provide to all agents to describe the formulation and the agent's objective in MDP when the model, i.e., the transition function and reward function, is unknown.

---

**Description of MDP with unknown model**

A finite horizon tabular Markov Decision Process (MDP) is a model for decision-making in scenarios where outcomes are influenced by both randomness and controlled decisions, with decisions being made over a finite number of time steps.

Components:

State Space $S$: $s_0, s_1, \ldots, s_{|S|-1}$, where $|S|$ is the total number of states.

Action Space $A$: $a_0, a_1, \ldots, a_{|A|-1}$, where $|A|$ is the total number of actions.

Transition probability matrix $P$: a three-dimensional tensor with shape $|S| \times |A| \times |A|$, where each entry represents the probability of transitioning from one state after taking a specific action to another state.

Reward matrix $R$: a matrix with shape $|S| \times |A|$, where each entry gives the mean of the immediate reward received after taking an action in a state.

Horizon length $H$: The total number of time steps the decision process is constrained to.

Number of episodes $K$: The total number episodes the MDP is repeatedly played by the agent, where in each episode, the agent starts fresh, makes a series of $H$ decisions and then the episode ends. Note that learning achieved in earlier episodes influences the behavior in later episodes. Unknown model of the environment: The transition probability

matrix $P$ and reward matrix $R$ are unknown to the agent, and the agent needs to estimate them based on the collected observations and improve its policy after each episode.

Interaction protocol:

For episode $k = 0, 1, 2, \ldots, K - 1$:

For time step $h = 0, 1, 2, \ldots, H - 1$:

Agent takes an action $a_{k,h} \in A$ based on the current state $s_{k,h}$

Agent receives reward $r_{k,h} := R[s_{k,h}, a_{k,h}] + \eta_{k,h}$, where $\eta_{k,h} \sim \mathcal{N}(0, 1)$

The environment transits to the next state $s_{k,h+1}$ with probability $P[s_{k,h}, a_{k,h}, s_{k,h+1}]$

Agent can update its estimation of matrix $P$ and $R$ based on the newly observed quadruples $(s_{k,h}, a_{k,h}, s_{k,h+1}, r_{k,h+1})$ for $h = 0, 1, 2, \ldots, H - 1$

Goal of the agent:

Maximize expected cumulative rewards $E\left[\sum_{k=0}^{K-1} \sum_{h=0}^{H-1} R[s_h, a_h]\right]$, where the expectation is w.r.t. randomness of agent's policy and state transition.

For STRIDE, since it can automatically update, store, and read the estimated transition and reward matrices in working memory, we simply use the following description about the problem instance for all episodes.

---

**Description of problem instance**

Now you are going to play in a finite-horizon tabular Markov decision process, with length of horizon {} (with time indices starting from h=0 to {}), number of states —S—={}, number of actions —A—={}. The transition matrix P and reward matrix R are unknown to you, so you need to estimate them based on interaction history.

---

For all the baselines, since they cannot reliably summarize the interaction history and construct the estimation of $P$ and $R$, we explicitly provide the estimation of $P$ and $R$ and the count of visitation of state-action pairs as shown below. This is similar to the "externally summarized interaction history" in the prompt for multi-armed bandit problems used by Krishnamurthy et al. (2024).

---

**Description of problem instance**

Now you are going to play in a finite-horizon tabular Markov decision process, with length of horizon {} (with time indices starting from h=0 to {}), number of states —S—={}, number of actions —A—={}. The transition matrix P and reward matrix R are unknown to you. Your current estimation of transition matrix P is {}, your current estimation of reward matrix R is {}, and your count of visitation of all the state-action pairs is {}.

---

## C.3. Dynamic Mechanism Design Problem

The following are the prompts we provide to all agents to describe the formulation and the agent's objective in Dynamic Mechanism Design problem, when the model, i.e., the transition function and reward function, is known.

---

**Description of dynamic mechanism design problem**

The dynamic mechanism design problem involves creating allocation and pricing rules for decision-making, where the value of resource to the agents changes over time as the state of the environment changes.

Components:

---

Players: a mechanism designer and a set of $N$ agents State Space $S$: $s_0, s_1, \ldots, s_{|S|-1}$, where $|S|$ is the total number of states.

Action Space $A$: $a_0, a_1, \ldots, a_{|A|-1}$, where $|A|$ is the total number of actions. Each action represents the mechanism designer's allocation of some scarce resource among $N$ agents.

Transition probability matrix $P$: a three-dimensional tensor with shape $|S| \times |A| \times |A|$, where each entry represents the probability of transitioning from one state after taking a specific action to another state.

Reward matrix $R$: a three-dimensional tensor with shape $N \times |S| \times |A|$, where each matrix $R[i, :, :]$ represents the reward matrix of an agent $i$ for $i = 1, 2, \ldots, N$, and each of its entry gives the mean of the immediate reward received by agent $i$ after the mechanism designer takes an action in a state.

Horizon length $H$: The total number of time steps the decision process is constrained to.

Interaction protocol:

Before the interaction starts, each agent $i$ reports a reward matrix (can be different from its true reward matrix $R[i, :, :]$), denoted as $\widetilde{R}[i, :, :]$, to the designer. Based on agents' reported reward matrix, the designer chooses a policy $\pi : S \to \Delta(A)$ and prices $\{p_i\}_{i=1}^N$ to be charged to each agent.

For time step $h = 1, 2, \ldots, H$:

Mechanism designer takes an action $a_h \sim \pi(s_h)$ based on the policy $\pi$ and the current state $s_h$

Each agent $i$ receives reward $R[i, s_h, a_h]$ for $i = 1, 2 \ldots, N$ The environment transits to the next state $s_{h+1}$ with probability $P[s_h, a_h, s_{h+1}]$

After the interaction, the mechanism designer charges each agent $i$ with some price $p_i$

Goal of the agents:

Each agent wants to maximize its utility $u_i = \mathbb{E}\left[\sum_{h=1}^H R[i, s_h, a_h]\right] - p_i$, that is, the difference

between the expected cumulative rewards, where the expectation is w.r.t. randomness of designer's policy and state transition, and the price charged by the mechanism designer. As the agents cannot directly take actions, their only leverage is to decide whether to truthfully report their reward matrix to the designer.

Goal of the mechanism designer:

Maximize the expected cumulative rewards of all agents $E\left[\sum_{i=1}^N \sum_{h=1}^H R[i, s_h, a_h]\right]$, where the expectation is w.r.t. randomness of designer's policy and state transition. As the designer only observes agents' reported reward matrix $\widetilde{R}$, to fulfil its objective, the designer needs to guarantee, with its policy and pricing strategy, no agent $i$ has incentive to report $\widetilde{R}[i, :, :]$ that is different from the true reward matrix $R[i, :, :]$ unilaterally.

It is known that VCG mechanism guarantees truthfulnes of the agents, and uniquely maximizes the objective. It is defined as follows:

$$\pi^\star = \arg\max_\pi \mathbb{E}_{\pi, P}\left[\sum_{i=1}^N \sum_{h=1}^H \widetilde{R}[i, s_h, a_h]\right]$$

$$p_i^\star = \mathbb{E}_{\pi_{-i}^\star, P}\left[\sum_{j \neq i} \sum_{h=1}^H \widetilde{R}[j, s_h, a_h]\right] - \mathbb{E}_{\pi^\star, P}\left[\sum_{j \neq i} \sum_{h=1}^H \widetilde{R}[j, s_h, a_h]\right]$$

for $i = 1, 2, \ldots, N$, where $\pi_{-i}^\star = \arg\max_\pi \mathbb{E}_{\pi, P}\left[\sum_{j \neq i} \sum_{h=1}^H \widetilde{R}[j, s_h, a_h]\right]$ is the optimal policy for a MDP with transition probability matrix P and reward matrix $\sum_{j \neq i} \widetilde{R}[j, :, :]$, that is, excluding the reward matrix of agent $i$ itself.

Now as a strategic decision maker, your job is to compute the VCG mechanism based on the given transition probability matrix $P$ and the reward matrix $R$ reported by the agents. Then you should take an action at each time step and charges prices to each agent at the end, according to your computed VCG mechanism.

> **Description of problem instance**
>
> Now you are going to play in a finite-horizon dynamic mechanism design problem, with number of agents N={}, length of horizon {} (with time indices starting from h=0 to {}), number of states —S—={}, number of actions —A—={}. The transition matrix P is:{} and reward matrix R reported by the agents is {}.

### C.4. Single-Issue Bargaining under Complete Information

The following are the prompts we provide to all agents to describe the formulation and the agent's objective in single-issue bargaining under complete information.

> **Description of single-issue bargaining under complete information**
>
> The alternating offer bargaining game is a negotiation framework between two players, a buyer and a seller, aimed at determining the price of an item. This strategic game plays out over several rounds with a finite deadline, emphasizing the tactics of bargaining under time constraints.
>
> Components:
>
> Players: Two (Buyer and Seller).
>
> Buyer's Value: 1 (the maximum price the buyer is willing to pay). Seller's Value: 0 (the minimum price the seller is willing to accept).
>
> Discount Factors ($\delta_b$ and $\delta_s$): Represents how much each player values immediate transactions over future possibilities, where $\delta_b, \delta_s \in (0, 1)$. Utility associated with future offers are discounted by $\delta_b^{t-1}$ and $\delta_s^{t-1}$ for the buyer and the seller, respectively, where t indicates the current round.
>
> Buyer's Utility: If a price $p$ is agreed upon at time step $t <= T$, then buyer's utility is $u_b = (1 - p) * \delta_b^{t-1}$.
>
> Seller's Utility: If a price $p$ is agreed upon at time step $t <= T$, then seller's utility is $u_b = (p - 0) * \delta_s^{t-1}$.
>
> Deadline: If no sale is agreed upon by the end of time T, the negotiation fails, and no transaction occurs, in which case, both agents get 0 utility.
>
> Complete Information: All details about the item's value range, the structure of the rounds, and the potential outcomes are common knowledge.
>
> Interaction Protocol:
>
> Decision Turns: Starting with the buyer, players alternate in making price offers. The player making an offer proposes a price within the range from the seller's value to the buyer's value.
>
> Responses: The opponent can either accept the proposed price, resulting in a sale and the game ending, or reject the offer, in which case the negotiation advances to the next round.
>
> Goal of the agents:
>
> The seller aims to maximize the sale price while the buyer seeks to minimize it. Each agent's goal is to negotiate a price as close as possible to their value (1 for the seller, 0 for the buyer) while considering the risk of no agreement by the deadline.

> **Description of problem instance**
>
> # For buyer
>
> This is the beginning of a new game instance, where you will play as the buyer. Your discount factor $\delta_b$={}, seller's

discount factor $\delta_s$={}, and the deadline T={}. In the following, you should make your decision by assuming your opponent is rational as well.

# For seller

This is the beginning of a new game instance, where you will play as the seller. Your discount factor $\delta_s$={}, buyer's discount factor $\delta_b$={}, and the deadline T={}. In the following, you should make your decision by assuming your opponent is rational as well.

**C.5. Single-Issue Bargaining under Incomplete Information**

The following are the prompts we provide to all agents to describe the formulation and the agent's objective in single-issue bargaining under incomplete information.

---

**Description of single-issue bargaining under incomplete information**

This is a finite horizon bargaining game with one-sided uncertainty, in which the uninformed bargainer, the seller, makes all the offers and the informed bargainer, the buyer, can only decides to accept or reject the offer.

Components:

Players: Buyer (informed) and Seller (uninformed).

Buyer's Value: b (the maximum price the buyer is willing to pay).

Seller's Value: 0 (the minimum price the seller is willing to accept).

Discount Factors ($\delta_b$ and $\delta_s$): Represents how much each player values immediate transactions over future possibilities, where $\delta_b, \delta_s \in (0,1)$. Utility associated with future offers are discounted by $\delta_b^{t-1}$ and $\delta_s^{t-1}$ for the buyer and the seller, respectively, where t indicates the current time step.

Buyer's Utility: If a price $p$ is agreed upon at time step $t <= T$, then buyer's utility is $u_b = (b-p) * \delta_b^{t-1}$.

Seller's Utility: If a price $p$ is agreed upon at time step $t <= T$, then seller's utility is $u_b = (p-0) * \delta_s^{t-1}$.

Deadline: If no sale is agreed upon by the end of time T, the negotiation fails, and no transaction occurs, in which case, both agents get 0 utility.

Information Asymmetry: Buyer himself knows the true value of b, which is drawn from a known distribution $F(v)$ supported on $[0,1]$. We assume $F(v) = v$, i.e., Buyer's value $b$ is sampled from a uniform distribution. The seller does not know $b$ but knows the distribution $F(v)$.

Interaction Protocol:

Decision Turns: In each time step $t = 1, 2, \ldots, T$, it is always Seller who makes an offer $p_t$ within the range of [0,1].

Responses: Buyer can either accept the proposed price, resulting in a sale and the game ending, or reject the offer, in which case the negotiation advances to the next time step.

Goal of the agents:

Seller's Objective: Maximize their expected payoff over the horizon of the game without knowing the true value of $b$. The seller must strategically decide on the prices $p_t$ to offer in each time step, considering the declining number of opportunities to make a sale and the distribution of $b$ inferred from the buyer's responses.

Buyer's Objective: Maximize their surplus, which is the difference between the true value $b$ and the price paid $p$, if a transaction occurs. The buyer needs to decide whether to accept or reject the seller's offers based on the value $b$ and the likelihood of a more favorable price in subsequent time steps, considering the finite number of time steps.

---

| Description of problem instance |
|---|

# For buyer

This is the beginning of a new game instance, where you will play as the buyer. Your discount factor $\delta_b$={}, seller's discount factor $\delta_s$={}, and the deadline T={}. Your value $b = \{\}$, which is uniformly sampled from $[0, 1]$. In the following, you should make your decision by assuming your opponent is rational as well.

# For seller

This is the beginning of a new game instance, where you will play as the seller. Your discount factor $\delta_s$={}, buyer's discount factor $\delta_b$={}, and the deadline T={}. The buyer's value $b$ is unknown to you, but you know it is uniformly sampled from $[0, 1]$. In the following, you should make your decision by assuming your opponent is rational as well.

**C.6. Tic-Tac-Toe**

The following are the prompts we provide to all agents to describe the formulation and the agent's objective for the Tic-Tac-Toe game. The prompts also detail the agents' goals and initial game setup.

| Description of Tic-Tac-Toe Game |
|---|

Tic-Tac-Toe is a classic two-player game where players take turns marking spaces in a 3x3 grid, aiming to place three of their marks in a horizontal, vertical, or diagonal row to win.
Components:
- Players: Two players, usually denoted as Player X and Player O.
- Board: A 3x3 grid where each cell can be empty, marked with an X, or marked with an O.
- Marks: Each player has a unique mark (X or O) that they place on the board.
Interaction Protocol:
- Players take turns starting with Player X.
- On each turn, a player marks an empty cell on the grid with their mark (X or O).
- The game continues until a player has three of their marks in a horizontal, vertical, or diagonal row, or all cells are filled resulting in a draw.
Rules:
1. Players alternate turns, with Player X always going first.
2. A player can only mark an empty cell.
3. The game ends when one player achieves a row of three marks horizontally, vertically, or diagonally, or when all cells are filled with no winner (a draw).
Goals of the Players:
- Player X: Maximize the chances of placing three X's in a row before Player O does.
- Player O: Maximize the chances of placing three O's in a row before Player X does.
Winning Conditions:
- A player wins if they place three of their marks in a horizontal, vertical, or diagonal row.
- If all cells are filled without any player achieving three marks in a row, the game results in a draw.
Game Setup:
1. The game begins with an empty 3x3 grid.
2. Players decide who will be Player X and who will be Player O.
3. Player X makes the first move.
Objective:
Each player aims to either achieve a row of three of their marks or to block the opponent from doing so. Strategic planning and anticipation of the opponent's moves are crucial to winning the game.

---

**Description of problem instance**

Now you are going to play a game of Tic-Tac-Toe. The current state of the board is {}. It is player {}'s turn. Your objective is to place three of your marks in a horizontal, vertical, or diagonal row to win while preventing your opponent from doing the same.

---

### C.7. Connect-N

The following are the prompts we provide to all agents to describe the formulation and the agent's objective for the Connect-N game. The prompts also detail the agents' goals and initial game setup.

---

**Description of Connect-N**

Connect-N is a generalized version of Connect-4, where two players alternate turns dropping colored discs into a vertically suspended grid. The objective is to form a horizontal, vertical, or diagonal line of $N$ discs. The game introduces a gravity effect where discs drop to the lowest available position within a column, adding a unique strategic dimension to the gameplay.

Components:
- Players: Two players, typically referred to as Player X and Player O, who use different colored discs.
- Board: A grid with configurable dimensions, larger than the typical 3×3 Tic-Tac-Toe board.
- Discs: Each player has an ample supply of discs in their respective colors.

Interaction Protocol:
- Players take turns, starting with Player X.
- On each turn, a player chooses a column to drop a disc into. The disc falls, affected by gravity, to the lowest available position within the column.
- The game continues until a player forms a line of N discs in a row (horizontally, vertically, or diagonally) or the board is completely filled, resulting in a draw.

Rules:
1. Players must alternate turns, with Player X always going first.
2. A player can only choose a column that has available space.
3. The game ends when one player forms a line of N discs or when all columns are filled without any player achieving this, which results in a draw.

Goals of the Players:
- Player X: Strategize to connect N of their discs in a row vertically, horizontally, or diagonally before Player O.
- Player O: Similarly, aim to connect N of their discs in a row while blocking Player X's attempts.

Winning Conditions:
- A player wins by aligning N of their discs in a row in any direction.
- The game results in a draw if the entire board is filled without either player achieving N in a row.

Game Setup:
1. The game starts with an empty board of the chosen dimensions.
2. Players decide who will play first (Player X) and choose their disc colors.
3. Player X makes the first move by dropping a disc into one of the columns.

Objective:
Each player aims to strategically drop their discs to form a line of $N$ while preventing their opponent from doing the same. Anticipating the opponent's moves and effectively using the gravity-affected game-play are critical to securing a victory.

---

**Description of problem instance**

> Now, you are going to play a game of Connect-N, where two players alternate turns dropping discs into a vertically suspended grid. The objective is to form a line of $N$ discs in a row, either horizontally, vertically, or diagonally. The current state of the board is {}, the current player is Player {}, the number of discs required to win is {}. Your objective is to strategically drop your discs to form a line of {} discs while preventing your opponent from doing the same.

## D. Additional Experiments

In this section, we conduct additional experiments that evaluate STRIDE and the baselines (GPT-3.5-Turbo-0125 with the temperature set to 0) on Tic-Tac-Toe and Connect-N Games. For these two games, we provide STRIDE with tools and demonstration that make it emulate Minimax algorithm as shown in Algorithm 10.

### D.1. Tic-Tac-Toe

**Agent's Objective in Tic-Tac-Toe.** The primary objective for each agent is to win the Tic-Tac-Toe game by placing three markers in the same row, column, or diagonal before the opponent. If a win is not feasible, the secondary objective is to aim for a tie, preventing the opponent from winning. Each agent strives to select the optimal action based on the game's current state. If both players play optimally, the game results in a tie.

**Experiment Setup and Results.** In addition to the baselines mentioned in Section 3, here we also include *RAFA with Monte Carlo Tree Search* (MCTS) (Liu et al., 2023) and *RAFA with Minimax*. For *CoT w/ code*, the LLM has been instructed to implement Minimax algorithm to play the game, and for the *RAFA* agents, the search breadth, denoted $B$, is set to 4. In addition to the original *RAFA MCTS* implementation[3], we implemented *RAFA with Minimax* as an extra baseline. We adopt the memory structure from their original implementation to store optimal actions and use similar prompts and interactions with the LLM to expand the game tree and assess game states. Additionally, for *RAFA with Minimax*, we set the search depth, denoted $U$, to the maximum value 9.

In our experiments, STRIDE is equipped with operational tools to emulate a Breadth-First version of Minimax algorithm with alpha-beta pruning (see Algorithm 10). Starting from depth 0 and progressing to the maximum depth — determined by the total number of empty cells on the board — the algorithm evaluates potential outcomes at each node: $+1$ for a win, $-1$ for a loss, and 0 for a tie or non-terminal states. Utilizing backward induction, the algorithm recursively refines and updates these scores, ensuring that the decision path optimizes the expected outcome at each node from the current player's perspective. These scores are stored in STRIDE's working memory. When STRIDE agent starts playing the game, it retrieves the scores for each possible action, and then selects the action with maximal or minimal score depending on the role of the player. We repeat the experiments on a fixed set of parameters for 10 runs, with the initial player being 'X' and an empty board to start the game. The results are presented in Table 6.

*Table 6.* Model performances in Tic-Tac-Toe (10 runs).

| Outcome | RAFA w/ Minimax | RAFA w/ MCTS | zero-shot CoT | zero-shot CoT w/ code | STRIDE |
|---|---|---|---|---|---|
| X Wins (%) | 50 | 60 | 70 | 80 | 20 |
| Tie (%) | 30 | 20 | 0 | 20 | **80** |
| O Wins (%) | 20 | 20 | 30 | 0 | 0 |

**STRIDE Vs. Baseline Models**  We also conducted experiments that pit STRIDE against baseline models in Tic-Tac-Toe, including *zero-shot CoT*, *zero-shot CoT w/ code*, and *RAFA w/ MCTS*. We instructed *zero-shot CoT w/ code* to implement the Minimax algorithm, and for *RAFA w/ MCTS*, we set $B = 4$ and $U = 4$. The experiments were conducted over 10 runs, with STRIDE playing as player 'X' and the baseline models as player 'O'. The outcomes are summarized in Table 7.

### D.2. Connect-N

**Agent's Objective in Connect-N.** In Connect-N, available moves can be made in the lowest empty space of each column. The agent aims to drop its discs to form a line of $N$ while preventing its opponent from doing the same. Each agent attempts to choose the best possible action based on the game's state. Similar to Tic-Tac-Toe, the game ends with a draw if both players

---

[3]https://github.com/agentification/RAFA_code

*Table 7.* STRIDE against Baseline Models in Tic-Tac-Toe (10 runs)

| Matchup | STRIDE Wins (%) | Tie (%) | Opponent Wins (%) |
|---|---|---|---|
| STRIDE vs zero-shot CoT | **90** | 10 | 0 |
| STRIDE vs zero-shot CoT w/ code | **80** | 20 | 0 |
| STRIDE vs RAFA w/ MCTS | **50** | 50 | 0 |

play optimally.

**Experiment Setup and Results** We conduct experiments with two configurations: (1) Connect-3 on a $3 \times 3$ board and (2) Connect-4 on a $4 \times 4$ board. Similar to the Tic-Tac-Toe game, STRIDE simulates the Breadth-First Minimax algorithm with pruning (see Algorithm 10) to find the optimal action in Connect-N. It first simulates every possible move and scores each node at each game's depth (1 for a win, -1 for a loss, and 0 for a tie or non-leaf node), then uses backward induction to update the scores for each game state. Using its working memory, STRIDE stores the computed scores for all possible actions at various depths. When the game starts, it selects the best action based on the computed scores. The results (averaged over 10 runs) are summarized in Tables 8 and 9.

*Table 8.* Model performances in Connect-3 (10 runs).

| Outcome | zero-shot CoT | zero-shot CoT w/ code | STRIDE |
|---|---|---|---|
| X Wins (%) | 60 | 90 | 30 |
| Tie (%) | 40 | 0 | **70** |
| O Wins (%) | 0 | 10 | 0 |

*Table 9.* Model performances in Connect-4 (10 runs).

| Outcome | zero-shot CoT | zero-shot CoT w/ code | STRIDE |
|---|---|---|---|
| X Wins (%) | 50 | 80 | 50 |
| Tie (%) | 10 | 0 | **50** |
| O Wins (%) | 40 | 20 | 0 |

We provide the following operational tools to STRIDE to help it emulate Algorithm 10:

- `CalculateScores`: expand every action at each depth and calculate the score for the nodes.
- `GetScores`: retrieve the computed scores for all the actions at the specified depth of the game tree.

---

**Algorithm 10** BFS Minimax with Alpha-Beta Pruning

---

1: **function** BFSALPHABETA($root$, $\alpha$, $\beta$)
2:     $queue \leftarrow$ new Queue()
3:     $parentMap \leftarrow$ new Dictionary()                    ▷ To store parent-child relationships
4:     $queue$.enqueue($\{root, \alpha, \beta\}$)
5:     $scores \leftarrow$ new Dictionary()                      ▷ To store scores temporarily
6:     **while** $queue$ is not empty **do**
7:         $\{node, current\_alpha, current\_beta\} \leftarrow queue$.dequeue()
8:         **if** $node$ is a terminal state **then**
9:             $scores[node] \leftarrow U(node)$                 ▷ Utility of terminal state
10:        **else**
11:            $value \leftarrow -\infty$ if $node$.isMaximizingPlayer() else $\infty$
12:            **for all** $child \in$ Children($node$) **do**
13:                $queue$.enqueue($\{child, current\_alpha, current\_beta\}$)
14:                $parentMap[child] \leftarrow node$
15:        **if** node in $parentMap$ **then**
16:            $parent \leftarrow parentMap[node]$
17:            $eval \leftarrow scores[node]$
18:            **if** $parent$.isMaximizingPlayer() **then**
19:                $scores[parent] \leftarrow \max(scores[parent], eval)$
20:                $current\_alpha \leftarrow \max(current\_alpha, scores[parent])$
21:            **else**
22:                $scores[parent] \leftarrow \min(scores[parent], eval)$
23:                $current\_beta \leftarrow \min(current\_beta, scores[parent])$
24:            **if** $current\_beta \leq current\_alpha$ **then**
25:                **break**                                     ▷ Pruning
26:     **return** $scores[root]$

---

