# OpenReview forum: "STRIDE: A Tool-Assisted LLM Agent Framework for Strategic and Interactive Decision-Making"
_ICML.cc/2024/Workshop/Agentic_Markets — Agentic Markets @ ICML'24 Poster_

### Official Review · Reviewer_jsGh · 2024-06-08
**Good paper on a suitable topic**

**Rating:** 7
**Confidence:** 3

**Review:**

This paper considers techniques for addressing difficulties in using LLMs in strategic decision-making settings, which are designed to address shortcomings of LLMs such as poor mathematical reasoning abilities, long-term planning, and anticipation of opponents' actions. The paper is well-written and experimental results cover a variety of settings. As far as I see this is absolutely a suitable topic for this workshop, and the techniques and results presented will be of interest to the participants. I therefore recommend accept.

---

### Official Review · Reviewer_LG81 · 2024-06-12
**Review on STRIDE: A Tool-Assisted LLM Agent Framework for Strategic and Interactive Decision-Making**

**Rating:** 7
**Confidence:** 4

**Review:**

## Review Summary
The paper introduces **STRIDE**, a framework designed to enhance the strategic decision-making capabilities of large language models (LLMs) such as GPT-4. The framework incorporates memory and specialized tools to assist LLMs in complex, multi-agent environments. Key contributions include:

1. **Development of the STRIDE Framework:** It integrates memory and specialized tools with LLMs to improve strategic reasoning in contexts like Markov Decision Processes, dynamic mechanism design, and bilateral bargaining games.
2. **Quantitative Evaluation:** The framework is empirically tested across various strategic settings, demonstrating improvements in decision-making capabilities.
3. **Implementation of a Structured Thought Process:** STRIDE employs a structured operation sequence and external memory to overcome LLM limitations in long-term planning and strategy.
---
## Position within the State of the Art
STRIDE advances the state of the art in strategic reasoning with LLMs, addressing gaps identified in existing research that primarily uses minimal input or chain-of-thought examples in strategic settings like matrix games and bargaining scenarios. Current studies often highlight the inconsistency and superficial understanding of game dynamics in LLMs (Davidson et al., 2024; Gandhi et al., 2023). Unlike approaches that integrate LLMs into fixed algorithmic behaviours like Monte Carlo tree search, STRIDE offers a flexible framework that can adapt various strategic behaviours using specialized tools and memory. However, its complexity and the heavy reliance on detailed, pre-defined operational setups could limit its applicability in dynamic real-world scenarios where adaptability and minimal configuration are crucial which could hinder its broader adoption and restrict effectiveness outside controlled experimental settings.

Note: The related work has been moved to Appendix which is not in compliance with the rules (?).

---
## Relevance to the Agentic Markets Workshop
- STRIDE's enhancements to LLM strategic decision-making capabilities, has relevance with the workshop’s focus on multi-agent market design and AI agents in economic models.
---
## Pros
- Significantly enhances LLM capabilities in strategic interactions.
- Robust testing across diverse settings confirms its effectiveness.
- Novel use of tools and memory to support LLM reasoning.

## Cons
- The framework's complexity could hinder its applicability and scalability without substantial customization.
- Its effectiveness seems to heavily rely on precise problem setups, hence limiting flexibility.
- The framework's reliance on detailed, pre-defined operational tools may not translate well to real-world scenarios where conditions and parameters are more fluid and less predictable.
- Given its dependency on training data and predefined tools, there is a risk that STRIDE could perpetuate existing biases in decision-making processes, leading to skewed or unfair strategic outcomes.


While STRIDE is a promising development in strategic decision-making for LLMs, its practical application faces significant challenges related to complexity, generalizability, and potential biases, which could limit its utility in dynamic and diverse real-world settings. Within the context of enhancing AI agents in strategic decision-making, it could expand on how these improvements impact the autonomy and reliability of AI agents.

General comments: The structure and organization of the paper is unclear. And much of the key material is stuffed in the appendices. In terms of scalability how is this statement: 'STRIDE can easily handle a state space of size 120' motivated? Fig 1 and Fig 2 are not truly connecting.

---